# Gut Microbiome Studies in Livestock: Achievements, Challenges, and Perspectives

**DOI:** 10.3390/ani12233375

**Published:** 2022-11-30

**Authors:** Giovanni Forcina, Lucía Pérez-Pardal, Júlio Carvalheira, Albano Beja-Pereira

**Affiliations:** 1CIBIO, Centro de Investigação em Biodiversidade e Recursos Genéticos, *InBIO* Laboratório Associado, Campus de Vairão, Universidade do Porto, 4485-661 Vairão, Portugal; 2BIOPOLIS Program in Genomics, Biodiversity and Land Planning, CIBIO, Campus de Vairão, Universidade do Porto, 4485-661 Vairão, Portugal; 3Universidad de Alcalá, Global Change Ecology and Evolution Research Group (GloCEE), Departamento de Ciencias de la Vida, 28805 Alcalá de Henares, Spain; 4Abel Salazar Institute of Biomedical Sciences, University of Porto, Rua de Jorge Viterbo Ferreira 228, 4050-313 Porto, Portugal; 5DGAOT, Faculty of Sciences, Universidade do Porto, Rua Campo Alegre 687, 4169-007 Porto, Portugal; 6Sustainable Agrifood Production Research Centre (GreenUPorto), Universidade do Porto, Rua da Agrária 747, 4485-646 Vairão, Portugal

**Keywords:** amplicon sequencing, genetic resources, metabarcoding, microbiota, next-generation sequencing, climate resilience, shotgun sequencing

## Abstract

**Simple Summary:**

The relentless capacity of sequencing every bit of DNA at low cost has been fueling major advances in several research areas. This also applies to the animal sciences, which witnessed unprecedented progresses in fields such as animal nutrition, health, and breeding. Particular attention has been paid to the gut microbiome, the community of microorganisms inhabiting the digestive tract of livestock species, and unforeseen developments have arisen. Nonetheless, such efforts have not been equal for the different livestock species, and the vast majority rely on widely-used standard techniques through which taxonomically useful genetic data are generated rather than more informative—yet computationally demanding—organismal genome-wide variation data. This review offers a glimpse of the gut microbiome research on five emblematic livestock species touching on the limitations regarding (i) the major methodological frameworks, (ii) species or breed, (iii) and spatial reach of these studies, thus providing valuable indications to fill current knowledge gaps and hopefully lay the basis for the planning of concerted research efforts. In this respect, we conclude that future studies should extend shotgun sequencing and transcriptomic approaches primarily to largely neglected ovicaprine and chicken breeds from rural areas of developing countries and microbial groups other than bacteria.

**Abstract:**

The variety and makeup of the gut microbiome are frequently regarded as the primary determinants of health and production performances in domestic animals. High-throughput DNA/RNA sequencing techniques (NGS) have recently gained popularity and permitted previously unheard-of advancements in the study of gut microbiota, particularly for determining the taxonomic composition of such complex communities. Here, we summarize the existing body of knowledge on livestock gut microbiome, discuss the state-of-the-art in sequencing techniques, and offer predictions for next research. We found that the enormous volumes of available data are biased toward a small number of globally distributed and carefully chosen varieties, while local breeds (or populations) are frequently overlooked despite their demonstrated resistance to harsh environmental circumstances. Furthermore, the bulk of this research has mostly focused on bacteria, whereas other microbial components such as protists, fungi, and viruses have received far less attention. The majority of these data were gathered utilizing traditional metabarcoding techniques that taxonomically identify the gut microbiota by analyzing small portions of their genome (less than 1000 base pairs). However, to extend the coverage of microbial genomes for a more precise and thorough characterization of microbial communities, a variety of increasingly practical and economical shotgun techniques are currently available.

## 1. Introduction

The massive decrease in sequencing costs associated with the generalization of high-throughput or next-generation sequencing (NGS) techniques has enabled unprecedented advances in microbiome studies spanning throughout the life sciences fields [1]. The strong bond between public health and the economy has been propelling the interest in microbiome research, which is deemed to hold a huge applicative potential under the One Health strategy [2] and other similar initiatives. Most of such studies addressing health and animal production have been mostly focused on gut microbiota, which is justified by the crucial role of these microorganisms in nutrition, fitness, and performance traits [3,4,5]. It is generally expected that advancing knowledge of the ruminant microbiome [6] bears a huge potential in terms of boosting animal production and health while lessening environmental pollution [7,8]. This promise seems of utmost importance when considering forecasts predicting an almost two-fold increase in the current production and consumption of meat in 30 years from now, with changes in dietary habits in developing countries—on top of human population growth—which will boost the demand for dairy products [9].

Some terms will be used extensively in this review and, for the benefit of the readers, their definition is provided as follows. The community of microorganisms inhabiting a given environment is referred to as a microbiota, while the term microbiome is used to indicate microbiota’s collective genomes [10]. On the other hand, the concept of metagenomics, first defined as “the direct genetic analysis of genomes contained in an environmental sample” [11], has been elaborated further as the “study of the structure and function of entire nucleotide sequences isolated and analyzed from all the organisms (typically microbes) in a bulk sample” [12].

However, the fast-increasing body of research produced in the wake of the above-mentioned compelling socioeconomic reasons has yet to cover much ground. Among the main shortfalls of this kind of study is the almost exclusive focus on cosmopolitan and highly selected breeds, which lacks representativeness both in terms of diversity and functionality, as the most promising knowledge may come from locally adapted native breeds. The role of microorganisms in the resilience and performance of livestock species is of paramount interest for their potential commercial value, especially in a time of rampant global change.

Another critical factor is the uneven attention being paid to bacteria, which include taxa that cause significant economic losses in addition to being a serious hazard to public health, e.g., [13]. On the other hand, the other microorganisms, such as fungi, archaea, protozoa [14], and viruses, have received far less attention. Yet another weakness is that the classical metabarcoding approach is still largely used, as opposed to increasingly feasible and affordable shotgun approaches that are now available (although only for a low number of samples) for a more precise and extensive characterization of microbial communities.

The goal of this review, which was prompted by the steadily increasing number of articles devoted to the livestock gut microbiome, is to assess the primary body of research in this subject, provide an overview of the state-of-the-art regarding sequencing approaches and knowledge produced, and then offer suggestions for future studies.

## 2. Next-Generation Sequencing Techniques

### 2.1. Amplicon Metabarcoding

This technique, known as the large-scale taxonomic identification of biological samples through the analysis of short DNA fragments of one or more genes (known as DNA barcodes), has benefited significantly from the development of high-throughput sequencing, which made it possible to process complex environmental samples [https://metazoogene.org/metabarcoding (accessed on 10 September 2022)].

According to the taxonomic group being targeted, different barcodes are preferred. For example, the V2-V3 and V3-V4 *16S rRNA* hypervariable regions have being traditionally used for bacteria [15,16], the V4 and V9 *18S rRNA* hypervariable regions for protists [17], and the internal transcribed spacer (*ITS*) *rRNA* regions (*ITS1* and *ITS2*) for fungi [18,19]. This PCR-based method relies on a dual indexing mechanism to simultaneously process huge numbers of individual libraries (i.e., samples) covering many taxa at a low cost.

The operational taxonomic units (OTUs) characterized by means of these amplicon libraries, however, are an underrepresentation of the true microbial diversity in the community of interest because DNA barcoding has lower sensitivity and limited resolution when compared to metagenomic data (i.e., spanning entire genomes). Instead, PCR and sequencing errors may result in its overrepresentation [20]. Nevertheless, since amplicon metabarcoding has been widely used by the scientific community worldwide for almost 20 years, a large number of homologous sequences are available for free download from GenBank as well as from other widely used public repositories, including Greengenes v13_8 [21], SILVA 138 [22], and RDP18 (Ribosomal Database Project) [23].

However, it is important to note that these databases are not always regularly updated (the most recent updates were made in 2013, 2019, and 2020, respectively), which can be problematic for users. Another reason to rely on this locus over others is the accessibility of user-friendly software such as the Quantitative Insights into Microbial Ecology pipeline (QIIME) [24], which implements *16S*-based tools for taxonomic assignment. This even inspired the development of software that predicts functional profiles of bacterial populations based on their *16S* sequences, such as Tax4Fun [25]. Overall, this locus has been used in the majority of livestock microbiome investigations carried out until now, necessitating the establishment of recommendations and best practices for the benefit of the animal science community as a whole [26].

### 2.2. Shotgun Sequencing

This technique entails randomly shearing one or multiple genomes (for instance, in the case of an environmental sample) into small DNA fragments that are then individually sequenced, mapped to reference genomes and then reassembled in the proper order (for a quick explanation, see [27]). Such a technique, which is not based on PCR, has the benefit of avoiding the formation of amplification artifacts and, by being not reliant on taxon-specific primers, may produce more thorough and reliable results in terms of the overall microbial diversity associated with a given sample thanks to its high sensitivity and resolution power.

The provision of knowledge on the biological functions encoded by the genome(s) being sequenced is another significant benefit [28]. However, because of its high sequencing costs, this technique is not yet scalable for large-scale surveys based on high numbers of samples, despite being simple and fast to execute [29]. The difficulties in recreating the microbial composition in the case of complex and large communities and the high computing expenses connected with data storage and processing are additional limitations of this technique [28,29]. The rapidly expanding community of scientists using shotgun sequencing is fortunate to have access to powerful bioinformatics tools that are being made available, with some like BLAST+ [30] allowing the buildup of customized reference databases based on the inclusion of freely available nucleotide and protein sequences from public repositories.

### 2.3. Metatranscriptomics

Referred to as the study of genes that are transcribed in microbial communities at a given moment and under certain environmental conditions as measured by the abundance of collective RNA transcripts [31], this culture-independent approach has delivered major insights into niche-specific transcript expression patterns and the ecological functions of microbial taxa within their community [32]. Overall, a common drawback of this suite of techniques, especially RNASeq, is the high costs, which are nonetheless expected to drop in the coming years in parallel with an increase in computational power and specific software such as HISAT [33] or ABioTrans [34].

## 3. The Significance of Microbiome Studies in Livestock Species

Only five animal species—cows, chickens, goats, pigs, and sheep—produce the great majority of the animal products that humans consume (meat, milk, eggs) [35,36,37]. Each of these species has its own evolutionary history that can be very deep, as is the case of the chicken—a bird—when compared with the other four, which are mammals. Even among the latter, there are notable differences not only in their digestive tracts, but also in terms of physiological aspects such as growth and lifespan as well as reproduction and behavior [38]. There are significant differences between ruminants, which possess a multi-chambered stomach (consisting of reticulum, rumen, omasum and abomasum) used to digest plant materials through fermentation, and monogastric animals, whose stomach is a simple structure made of a single compartment. The advantages of the ruminants regarding their capacity to obtain energy from poor-quality food and the limitations experienced in maintaining a balanced and healthy ruminal flora do not apply to the monogastrics, which are characterized by a faster development and a shorter lifespan.

Indeed, the digestive tract is the most structural factor in animal production as its functionality and health determine most of the individual’s performance [39], and is therefore the region that has been attracting the vast majority of microbiome research, followed by the reproductive tract [40]. Yet, more recently, environmental and public health concerns have also been the focus of a growing number of these studies, namely regarding greenhouse gas emissions [41], the spread of food-borne pathogens [42], and the rise of antibiotic resistance [43].

## 4. Microbiome Studies in Livestock Species

### 4.1. Ruminants

#### 4.1.1. Cattle

The majority of gut microbiome studies in cattle have focused on the characterization of microbial communities by *16S rRNA* gene amplicon sequencing as a consequence of different animal diet composition [44,45], gastrointestinal tract (GIT) location [46], feed efficiency [47], breed-specificity [48], metabolic disturbs [49], changes over time [50] and individual specificities [51,52], as well as across housing types and farms [53]. Interestingly, special attention has been devoted to identifying individual-based differences irrespective of age, sex, breed, or environment [54], with patterns of similarity and dissimilarities helping to define the core microbiome in the bovine rumen [51] as well as other livestock [55]. At the same time, it was evidenced that differences in taxonomic composition and the underlying community metabolic networks may still result in functional similarity [56], as well as that the metabolic potential of the rumen microbiome may be diet-driven [57]. A large-scale survey of dairy cows indicated that the core rumen microbiome composition underlies not only animal productivity but also the nature of their emissions [58]. It was only recently that shotgun metagenomics opened the door to a thorough exploration of the rumen microbiome composition in cattle, enabling the assembly of entire bacterial genomes (most of which belong to new taxa), and the identification of new enzymes [59]. This approach has also allowed for the elucidation of the interplay between the rumen microbiome along with its metabolome and the host metabolome, shedding new light on the finest mechanisms underlying production performances in dairy cows [60].

#### 4.1.2. Cattle Microbiome Profiling

Compared to other ruminant livestock species, the cattle gut microbiome is probably the one that has been explored more intensively, which provides an exhaustive picture of the bacterial communities inhabiting different GIT locations. The most abundant phyla are represented by Bacteroidetes and Firmicutes, which may account for more than 90% of the entire GIT bacterial community, with Actinobacteria, Proteobacteria, Spirochaetes, and Tenericutes representing other major yet comparatively less abundant taxa [49,53,61,62,63,64,65]. Bacteroidetes and Firmicutes are dominated, respectively, by classes Bacterioidia and Clostridia along with Bacilli. Concerning the major orders (Figure 1), the former class mostly consists of Bacteroidales, while the latter one of Clostridiales [59]. The most abundant families include Bacteroidaceae, Clostridiaceae, Lachnospiraceae, Peptostreptococcaceae, Rikenellaceae, and Ruminococcaceae [53,55], while dominant genera—not only in cattle but in adult ruminants as a whole [55]—are *Butyrivibrio*, *Prevotella* and *Ruminococcus* [51,60,63,64,66]. Genus *Clostridium* is also abundant in cattle rumen [65] along with *Acetitomaculum*, *Acinetobacter*, *Mogibacterium*, *Succiniclasticum*, and *Treponema* [46]. Based on recent studies, genera like *Fibrobacter* and *Ruminococcus* are among the core heritable bacteria transferred vertically across generations in the light of their primary role in cellulolysis [58]. A detailed list of the GIT-associated bacterial taxa and the pertinent bibliographic references in cattle is reported in Appendix A.

#### 4.1.3. Sheep

The last decade has witnessed a mounting interest in sheep microbiome research. A recent study based on bacterial *16S* has confirmed that, similar to what was found in cows, the microbial hosts may be responsible for alterations in terms of feed efficiency [67], while other works have suggested that feeding strategies may promote a more or less diverse microbial community [68,69]. Additionally, compositional changes in the microbiome have been observed along the GIT [70,71] and as an effect of parasite infections [72]. In sheep, however, the compositional changes of the archaeal rather than the eubacterial community play a main role in feed efficiency, with the latter exerting its main influence in terms of the presence/absence pattern of only a few specific taxa [67]. Another recent *16S* study compared the microbiome composition in sheep and goats, finding no substantial differences between the two taxa; however, variation did occur depending on age, with older individuals hosting a higher microbial diversity [73], similar to what has been found in Tibetan sheep [71]. Interestingly, differences in gut bacterial compositions have been observed among different Chinese sheep breeds from the Tibetan Plateau [74], contradicting what was found in a similar study on Italian sheep where microbiome differences were mostly due to different husbandry practices [75]. Like in cattle, however, feed efficiency turned out to be related to a higher abundance and diversity of rumen microbiomes [76]. Other studies have been carried out on local breeds of high socioeconomic relevance, often revealing a fairly diverse composition, as in the case of the Chinese Mongolian sheep [77], or, similar to what was found in cattle and goats [78], a marked heterogeneity across different GIT locations as in the case of the Qinghai semi-fine wool sheep [71]. Notably, some recent studies addressing a likely association between host genetics and rumen microbiota in local sheep breeds have unveiled the modulating effect of ovine candidate genes on its composition [79] and the interplay between this and host gene expression in maintaining homeostasis in extreme environments [80]. Nevertheless, all the previous studies are based on *16S* metabarcoding, while applications of shotgun metagenomics to characterize the gut microbial composition in sheep are still scant. In this respect, however, it is worth mentioning a study combining the two approaches with metaproteomics to explore the link between microbial communities and biochemical pathways [81].

#### 4.1.4. Sheep Microbiome Profiling

The characterization of the sheep GIT microbiome has revealed its substantial similarity in composition with that of cattle and other ruminants, with Bacteroidetes and Firmicutes making up more than 80 to 90 percent of the gut microbial community [67,69], followed by the phyla Actinobacteria, Proteobacteria, Spirochaetes, and Verrucomicrobia [68,74,82]. Bacterioidia and Clostridia are the dominant classes [75]. Moreover, Bacteroidales and Clostridiales figure among the most abundant orders (Figure 1), while, similar to what is observed in cattle, Eubacteriales and Lactobacillales stand out among Firmicutes. As far as the family-level is concerned, Ruminococcaceae and Lachnospiraceae emerge [74] along with Prevotellaceae, Rikenellaceae, and Succinivibrionaceae [67,76,80]. Concerning the most prevalent genera, *Prevotella* outstands [80], followed by *Acinetobacter* [79], *Campylobacter* [75], *Bacteroides*, *Desulfovibrio*, *Oscillospira*, *Ruminococcus*, *Treponema* [77], *Fibrobacter,* and *Succinivibrio* [76]. A detailed list of the GIT-associated bacterial taxa and the associated bibliographic references in sheep is provided in Appendix A.

#### 4.1.5. Goat

Molecular studies aimed at characterizing gut microbiome composition in this livestock species are still scarce in comparison to sheep, despite the economic relevance of goat meat and dairy products. Interesting exceptions, however, do occur, such as a work exploring the effects of dietary nitrate addition on microbial composition and ruminal fermentation based on a combined metabarcoding approach employing *16S* and *18S* amplicon libraries to characterize bacteria and protists along with fungi, respectively [83]. Other studies have evidenced the role played by fat acid supplementation [84] and a grain-rich diet [85] in shaping the bacterial and fungal diversity of rumen microbiome based on *16S* and *ITS* metabarcoding, respectively. Interestingly, a recent study based on amplicon libraries of the three loci mentioned before evidenced the role played by specific fungal and bacterial consortia in enabling lignocellulose breakdown by means of the production and interaction of a suite of specific metabolites [86]. Consistently, the *16S*-based methanogenic archaea diversity has turned out to be associated with a diet rich in condensed tannin-containing pine bark [87]. Current investigations have evidenced that, in goats as well, the microbial community varies throughout different GIT sectors [88] and tends to increase with age in young individuals [89,90], improving their productive performances [88,91,92]. Concordantly, the inoculation of rumen fluids during early life stages was found to boost the development of the rumen microbiome and even accelerate weaning [93], while the occurrence of apicomplexan parasites in goat kids was found to be associated with a decrease in the abundance of butyrate-producing bacteria, leading to an increase in mucosal inflammation and tissue repair [94]. Contrarily, it was discovered that antibiotic-induced gut microbiota dysbiosis likely worsened disease by encouraging inflammatory immune responses. [95].

Differences in the microbial composition have emerged when comparing adults belonging to different goat breeds [96], even if diet and environment seem to be the more important drivers of microbial diversity than genotype [97]. The occurrence of given bacterial hosts, in turn, was found to be associated with the digestibility of dietary phosphorus [98]. However, the exploration of gut microbiome components other than bacteria is quite limited in goats, with one of the few exceptions being represented by a study employing *16S* and *18S* amplicon libraries to explore the bacterial and ciliate protozoal diversity, respectively, in relation to the effects of antibacterial peptides on rumen fermentation function [99]. Moreover, the application of shotgun approaches to the characterization of the gut microbiome in goats is still limited to a single recent study [78].

#### 4.1.6. Goat Microbiome Profiling

Compared to other ruminant livestock species, goats are probably those that have so far received less attention concerning gut microbiome studies. Bacteroidetes and Firmicutes are the dominant bacterial phyla (i.e., accounting for more than 80% of the GIT bacterial community), followed by Proteobacteria [89,90,98] and Verrumicrobia [84,88] along with Fibrobacteres, Spirochaetes, and Tenericutes [73,85]. As far as the most abundant orders are concerned, Bacteroidales and Clostridiales—similar to what is observed in cows and sheep—prevail over others (Figure 1) [96]. The dominant families include Prevotellaceae, Veillonellaceae, and, to a lesser extent, Lachnospiraceae, Rikenellaceae, and Ruminococcaceae [84,98]. Among the dominant genera, *Prevotella* stands out along with *Bacteroides*, *Butyrivibrio*, *Clostridium*, *Oscillospira*, *Ruminococcus*, *Succiniclasticum*, and *Succinivibrio* [73,84,85,88,90,92,96,100]. A list of the GIT-associated bacterial taxa and the related literature in goat is reported in Appendix A.

### 4.2. Monogastric

#### 4.2.1. Pig

Microbiome research in the pig industry has been propelled by the need to reduce animal stress that may otherwise turn into economic losses for farmers [101]. In this context, weaning is a critical life stage in which the piglet diet undergoes a sharp change. Studies on the swine gut microbiome have largely benefited from the establishment of a reference gene catalogue by means of deep metagenome sequencing of fecal samples [102] and have confirmed that also in this livestock species the interplay between diet and gut physiology across different growth stages is intimately associated with animal health and production performance [103], including fat deposition [104]. Other than varying on the basis of the food provided [105,106,107,108], GIT location [109,110,111], behavior [112], parasite infections [113], breed affiliation, and sex [114], the microbial diversity was found to correlate positively with piglet weight [115] and age [116]. Likewise, studies combining *16S rRNA* metabarcoding and shotgun metagenomic sequencing revealed that the composition of the pig gut microbiome varies considerably and predictably across the lifespan [117]. This is particularly evident postweaning [118], when a higher microbial diversity underlies an increase in the genes associated with oxidative stress and heat shock compared to nursing piglets [119]. Interestingly, some studies evidenced that the combination of culturomics and shotgun metagenomics—an approach seldom applied to other livestock species—may deliver a more exhaustive picture of gut [120,121] and antimicrobial resistance [122]. Investigations based on the combination of *16S rRNA* metabarcoding and shotgun metagenomics have delivered insights into antimicrobial resistance dynamics in pig farms [108], while *18S rRNA* metabarcoding of fecal samples allowed to draw up a detailed list of intestinal protist parasites [123]. The combination of *18S* and *ITS* amplicon libraries has been recently used to characterize the pig gut microbial eukaryote community, finding the association of some of its members with host body weight [124], while that of *16S* amplicon data and metagenomics has delivered unprecedented insights into the functional and taxonomic diversity of the pig gut microbiome [123].

#### 4.2.2. Pig Microbiome Profiling

Notwithstanding the pronounced GIT structural differences between ruminants and monogastric animals such as pigs, the gut microbiome of the latter is also dominated by phyla Bacteroidetes and Firmicutes [115], followed by Proteobacteria [103,112], with Bacteroidia and Clostridia being the most abundant classes along with Bacilli [112,124]. Similar to what was observed in other livestock species, the dominant orders are Bacteroidales and Clostridiales (Figure 2), while the most abundant families are Bacteroidaceae, Enterobacteriaceae, Lachnospiraceae, Lactobacillaceae, Prevotellaceae, and Ruminococcaceae [106,108,116]. The genera most commonly found in the GIT of adult pigs are *Alloprevotella*, *Bacteroides*, *Escherichia*, *Lactobacillus,* and *Prevotella* [110,120,125,126] along with *Clostridium*, *Desulfovibrio*, *Enterococcus*, *Fusobacterium,* and *Streptococcus* [127,128,129]. A list of the GIT-associated bacterial taxa and the pertinent bibliographic references in pig is provided in Appendix A.

#### 4.2.3. Chicken

Microbiome research in chicken has made great strides since the advent of NGS techniques, as testified by the studies based on comparative metagenomic pyrosequencing to characterize the cecal microbiome in pathogen-free and infected individuals [130] and to explore the effect of antimicrobials on its communities as well as in relation to the abundance of antimicrobial resistance genes [131]. Nevertheless, most of these investigations are based on *16S rRNA* metabarcoding [132], while shotgun metagenomics is just taking its first steps in the poultry sector, with comparative studies applying the two approaches pointing to the much higher resolution power of the latter [133]. Shotgun metagenomics has also recently been employed to assess the role of dietary supplementation in improving the health status—and hence the productive performances—in broiler chickens by fostering the diversity of their cecum microbiome [134], also in the form of in ovo supplementation [92], as well as to characterize new bacterial, archaeal, and bacteriophage taxa of the chicken gut microbiome [135], thus shedding light on their biological function [136]. However, *16S rRNA* metabarcoding alone is still widely used to compare the microbiome composition of healthy versus unhealthy individuals as a consequence of viral, e.g., [137], or bacterial [138] infections, of individuals subjected to different dietary treatments [139], as well as across different indigenous breeds [140], GIT locations [141], rearing systems [142] and individual lifetimes [143], with a special focus on improving growth performance by transplanting cecal [144] or fecal [92] material between individuals of different age groups. However, compared to other livestock species, the non-bacterial component of the gut microbiome has been given more attention and most of the studies focus on possible pathogens such as *Cryptosporidium* [145].

#### 4.2.4. Chicken Microbiome Profiling

Similar to what occurs in the GIT of other livestock species, the most abundant microbial phyla in chicken are Bacteroidetes and Firmicutes [136,146], even though sometimes Proteobacteria are more abundant than the former [135,138,145], while Bacilli, Clostridia, and Gammaproteobacteria are the dominant classes [134]. At the order level, Bacillales, Enterobacteriales, Lactobacillales, and Campylobacterales are the most common groups (Figure 2), while the most prevalent families include Enterobacteriaceae and Lactobacillaceae [139]. As far as the dominant genera are concerned, *Alistipes*, *Bacteroides*, *Clostridium*, *Helicobacter*, *Lactobacillus*, and *Ruminococcus* [133,143,144,146,147,148,149] stand out as well as *Flavobacterium* [139], *Campylobacter,* and *Veillonella* [150]. A detailed list of the GIT-associated bacterial taxa and the related bibliographic references in pig is reported in Appendix A.

## 5. Resistome

The term “resistome” was introduced approximately two decades ago to indicate “the resistance determinants present in the soil” associated with bacterial populations living therein and showing multidrug resistance higher than expected [151]. The expression “bacterial resistome” has since become increasingly popular, while its meaning has evolved into the suite of all antibiotic resistance genes (ARGs) and their precursors in both pathogenic and nonpathogenic bacteria as well as antibiotic producers [152]. With a fast-growing body of research published on this topic, the concept of resistome has further evolved to incorporate different types of resistance and is now a key element in the framework of the One Health approach [153].

The identification of antimicrobial ARGs in bacteria inhabiting livestock GITs is crucial in animal science. An investigation specifically focused on the fecal bacterial resistome used a combination of the two approaches, traditional *16S* metabarcoding and shotgun metagenomics, evidencing the strong link between diet and antimicrobial resistance [154]. Moreover, the specificity of the microbial hosts in different GIT locations has emerged in a study on wild and domestic ungulates including cattle and goats [78]. This result serves as a model for future association research by highlighting the significance of local physiological changes along the GIT for various hosts. The advent of innovative nanopore technology, which enables large-scale research to highlight the most abundant resistance genes that may have a significant influence on animal, human, and environmental health, has spurred the rapidly expanding interest in the cow resistome [155]. On the other hand, studies on the bacterial resistome associated with the sheep gut microbiome are still scant when compared to those carried out in cattle or other livestock [156], and no specific investigation has been carried out on goats, but a recent study flagged as many as 30 ARGs in the sheep rumen, most of which related to daptomycin and colistin [157].

As far as non-ruminant livestock species are concerned, the scenario is even more complex. Pigs have received special attention in terms of characterization of the gut bacterial resistome, with recent studies demonstrating differential expression in humans, chickens, and specifically pigs [158]. Yet in chickens, the investigation into the ARGs associated with the gut microbiome has shown that the predominant classes are largely the same as those detected in pigs, including tetracycline, aminoglycoside and macrolide–lincosamide–streptogramin [159]. Of particular interest and utmost topicality is the risk of the potential transmission of ARGs from poultry meat to humans [160].

## 6. Metagenome and Functional Profile Prediction

Over the last years, a plethora of bioinformatics tools, including PICRUSt (Phylogenetic Investigation of Communities by Reconstruction of Unobserved States [161]), PICRUSt2 [162] along with FaproTax [163], the already mentioned Tax4Fun [25] and Tax4fun2 [164], have been made available to the scientific community for the purpose of predicting the functional profiles of the microbiota investigated in different studies. Moreover, Kyoto Encyclopedia of Genes and Genomes (KEGG) pathway analysis is often used in combination with these software to predict their metagenomic contributions. Since the vast majority of microbiome investigations have so far relied on *16S rRNA*, the algorithms of this type of program map the copies of this gene that were obtained in a given study to its homologs in the phylogenetically closest taxa with fully sequenced genomes. In other words, this approach allows predicting the functional metagenomic content without sequencing the entire genomes of the taxa which are actually present in the sample analyzed. Noteworthy, these software can work not only with amplicon metabarcoding but also shotgun sequencing data, even though their accuracy largely relies on available reference genomes and, as of now, it is still severely biased toward human datasets [165]. In addition, it is worth mentioning that a recent soil microbiome study comparing amplicon and shotgun sequencing functional profiling suggested that PICRUSt performs better than Tax4Fun to detect omnipresent functions, whereas Tax4Fun predicted greater abundances of functions from more specialized pathways [166]. Since the predictive tool used can lead to making different inferences, the authors suggested to reap the benefits of combining them rather than relying on either one or another [166].

The paucity of available reference genomes has been specifically invoked by some authors as the reason preventing them from performing functional prediction, e.g., [53], but others who nonetheless opted to perform it still detected significant differences in the predicted metagenomic profiles among GIT locations in dairy cows [46], sheep [77], goats [82], and pigs [126], thus pinpointing the most important metabolic pathways across different gastrointestinal microbial ecosystems. Additionally, functional prediction and KEGG analysis have been applied to unveil the differences in terms of metabolic pathways in the rumen of sheep cohorts with different feed efficiency [76] as well as in the same sheep sampled in different periods of the year [80], goats of different age [90], piglets with different body conditions [115], adult pigs of different breed and sex [71], and chickens of different age [167], breeds [140] or with a different health status [137]. It is conceivable that with the fast-growing increase in reference genomes available, metagenome and functional profile prediction tools will become more and more accurate in the next future, and their employment should be envisaged in any gut microbiome study.

## 7. Gut Microbiome, Health, and Welfare in Livestock

The positive or negative roles played by gut microbial taxa on health, welfare, behavior and performances in livestock exposed to different production conditions have been extensively discussed in several studies, e.g., [168,169,170,171,172,173,174]. It is worth touching on here why the studies addressing this topic are in such high demand nowadays and how crucial they are in animal farming as well as to broader society. In this respect, it is pertinent to mention their contribution to the development of non-antibiotic microbial therapies based on probiotics [175] as well as in pinpointing biomarkers of feed efficiency to deploy strategies that can notably improve livestock production performances [67,68,76,84,85,98,126,134,148,176] and growth [69,93,115]. Additionally, gut microbial profiling is paramount to monitor livestock health status and set up treatments to boost it, e.g., [113,125,139,144,165,177] as well as to prevent the establishment or aggravation of pathologic states [64,137,138,150] and evidence peculiar adaptations of local breeds to harsh environments [80].

## 8. Conclusions

The major advances in high-throughput sequencing technology have opened a new era in the study of the livestock gut microbiome, the composition and function of which are tightly associated with animal health and productive performance. The information produced has a profound social and economic impact. Previous attempts to take stock of available gut microbiome studies in livestock were mostly focused on cattle and chickens, or on the microbial groups rather than their hosts. In cattle, microbiome composition has been widely investigated in terms of feeding-related changes and their impacts on production strategies or environmental issues associated with ruminal methane emissions. We have expanded this review to other livestock animals, trying to make the point about what has been mostly performed so far and what is still lacking. A first consideration deals with the subjects of the microbiome studies carried out so far, in which priority was given to some species (such as cattle and pigs) rather than others (such as goats). Additionally, there is a clear bias in terms of the breeds investigated: expectedly, most studies are focused on a few cosmopolitan and highly selected breeds, while local breeds from rural areas are largely neglected, even though livestock research is a fundamental component to boost development strategies and the socioeconomic level of associated human communities. Characterizing the microbiome composition and its interaction with the host in non-intensive husbandry systems might, for instance, provide useful information on how to optimize livestock productivity through nutrient supplementation. Furthermore, it should be noted that a considerable number of microbiome studies, also on local breeds, have been carried out in China or Europe, while much less attention has been devoted to Africa and the tropical and subtropical regions as a whole.

Overall, the body of literature examined in this review allows us to conclude that livestock microbiome composition is affected by age, food, sex, and taxonomy, even if core bacteria occur across the GITs of different species. Admittedly, however, knowledge of other microbial groups is scant. As far as the methodological approach is concerned, shotgun metagenomics is still insipid when compared to amplicon metabarcoding sequencing, even though the few comparative studies employing both approaches on the same datasets evidenced the tremendously higher detection power of the former one, which is much more efficient in identifying underrepresented taxa whose detectability may be biased by the failure of universal primers to hybridize all templates as well as by its reliance on the number of hypervariable regions targeted, e.g., [133,178]. Conventional wisdom suggests that comparisons between studies based on either amplicon metabarcoding or shotgun sequencing on different datasets and with different experimental settings should not be made, but in general it can be stated that the latter is more reliable when estimating the absolute abundance of different microbial taxa. In addition, it is worth mentioning that in an increasing number of studies the two approaches are combined to first obtain a general picture of microbial diversity in the entire sample via amplicon metabarcoding and then, on this basis, select samples for shotgun sequencing to carry out functional analysis with higher accuracy, e.g., [108,127]. It is conceivable that with the fast-decreasing sequencing cost and the increasing suite of powerful bioinformatic tools available, the much more insightful shotgun sequencing will replace amplicon metabarcoding in most gut microbiome studies. This will presumably translate into expanding not only the taxonomic breadth and resolution but also the focus of the research. Indeed, bacteria is the most-studied microbial group compared to the others, which are most often given some attention only when represented by parasites of commercial relevance, but having a large amount of information encompassing other microbial groups as well may trigger interest in promoting research on them.

Indeed, focus on commensal protozoa, which nonetheless may still play a major role in regulating bacterial populations they feed on, is still limited, as is research on the fungal component of the gut microbiome. To achieve a comprehensive knowledge of the function of the microbiome and its underlying dynamics, the characterization of microbial groups other than bacteria is of key importance and should be addressed in future studies. As far as the NGS approach is concerned, it is important to note that the choice is often based on, other than budgetary issues, the availability and accessibility of comparative data as well as on the bioinformatics hurdles associated with the newest and most comprehensive techniques, which may prevent some research groups from applying them due to their limited computational resources and/or expertise. Enhancing the integration of metatranscriptomic studies—which are particularly scant for non-bacterial components—into microbiome research would allow a better understanding of the functional role of different microbial groups in the gastrointestinal tract. Having such valuable tools should not deter researchers from embracing more exhaustive approaches such as those based on shotgun genomics or metatranscriptomics, which, on the one hand, are less affordable and more computationally intensive, yet, on the other hand, may deliver much larger and more accurate amounts of information. These efforts are fully justifiable if we consider that a major application of genomic data in relation to livestock studies is on animal and human health, where epidemiological investigations are fueled by the prospect of threats to human activity and public health with great impact on state wealth. Moreover, the integration of metagenomics and metatranscriptomics with metabolomics and proteomics (multi-omics sequencing) could provide more valuable information about the interaction of the complex “host-microbiota-environment”, which could be useful for deploying future applications and interventions. In this context, there is a pressing need to better our understanding of the reciprocal influence of coexisting humans and livestock on each other’s gut microbiome and resistome.

## Figures and Tables

**Figure 1 animals-12-03375-f001:**
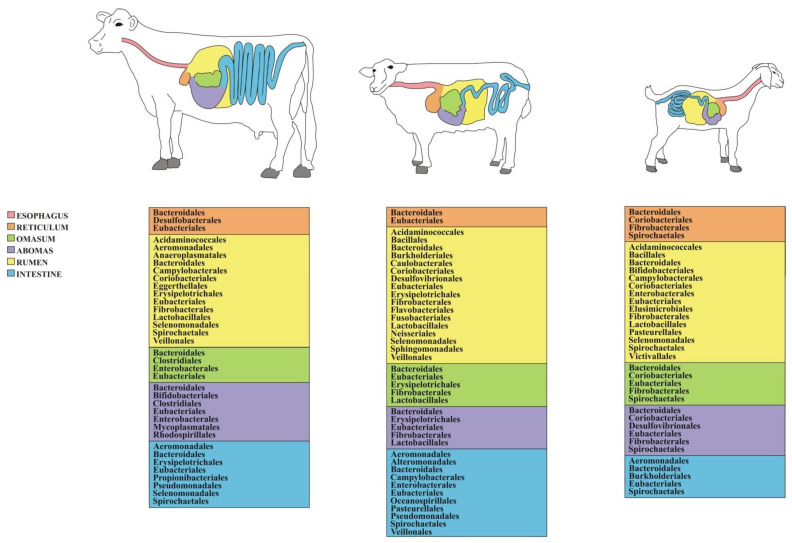
List of the most abundant bacterial orders found across the GITs of different ruminant livestock species (see Appendix A for further details). For the sake of clarity, the intestine designation may refer to both small and large intestines.

**Figure 2 animals-12-03375-f002:**
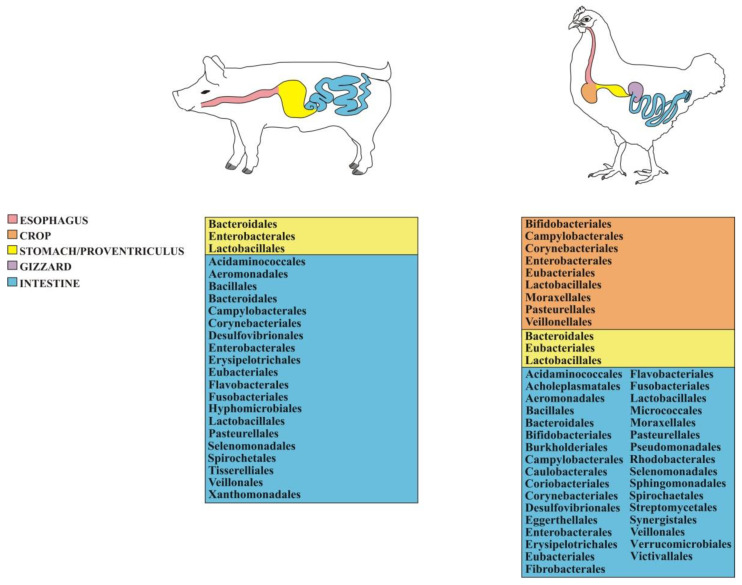
List of the most abundant bacterial orders found across the GITs of the two monogastric livestock species presented in this study (see Appendix A for further details).

## Data Availability

Not applicable.

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
