# Peer review of "Gut Microbiome Studies in Livestock: Achievements, Challenges, and Perspectives"

_animals, 2022, doi:10.3390/ani12233375_

Round 1

Reviewer 1 Report

Page 2/18; Line #84: Should the word 'bacteria' be deleted from this sentence.

Page 3/18; Lines #100-101:  The reference [15] does not deal with the bacteria species. Authors are advised to check if citation is relevant. This may be checked for other citations as well.

The conclusion section is very lengthy. It may be shortened appropriately to make it precise.

The authors are also advised to improve the resolution of Figures 1 and 2. The listing of the orders of the bacterial species is difficult to read.

Author Response

REFEREE #1

Page 2/18; Line #84: Should the word 'bacteria' be deleted from this sentence.

Our reply: Correct: as requested, this term was deleted.

Page 3/18; Lines #100-101:  The reference [15] does not deal with the bacteria species. Authors are advised to check if citation is relevant. This may be checked for other citations as well.

Our reply: We replaced this reference with another (Drancourt et al. 2000) that was deemed suitable. We also double-checked the relevance and correct order of all other citations in the MS.

The conclusion section is very lengthy. It may be shortened appropriately to make it precise.

Our reply: This section is longer than normal, but also because this is a review article, and not a research report, which includes a Discussion section followed, possibly, by the Conclusions. Logically, this review lacks a proper Discussion section, as it is the essence of a review paper. As a result, the Conclusions section of this MS is the only one where we can summarize the main points. Although the length is fine as it is, we did remove a few sentences to address some criticisms made by the other referees.

The authors are also advised to improve the resolution of Figures 1 and 2. The listing of the orders of the bacterial species is difficult to read

Our reply: We improved the resolution and font of Figures 1 and 2 as indicated.

Reviewer 2 Report

Gut microbiome studies in livestock: achievements, challenges 2 and perspectives. This is very interestng and informative review on the genetic information of microbiom in livestock. The review has accomulated important studies and have reported them in very good way. I think after careful revision, the paper must be accepted. 

I think the authors should provide informaiton in one or several tables. More emhasis should be given on future studies and strategies. Introduction seems to be very small if possible it can be expanded.

Also why some of the livestock species have been skipped. 

Some of the statements are without references, Proper references should be added to such sentences. 

Author Response

REFEREE #2

Gut microbiome studies in livestock: achievements, challenges 2 and perspectives. This is very interesting and informative review on the genetic information of microbiome in livestock. The review has accumulated important studies and have reported them in very good way. I think after careful revision, the paper must be accepted. 

We appreciate the reviewer's positive assessment, and we will try our best to address the reviewing panel's criticisms.

I think the authors should provide information in one or several tables. More emphasis should be given on future studies and strategies. Introduction seems to be very small if possible it can be expanded.

Our reply: Please, note that we have used Supplementary tables 1 and 2 to summarize information we deemed relevant, while we also had to stick with the space constraints associated with the format chosen – a review – that is limited to two display items (i.e., figures and/or tables).

Also why some of the livestock species have been skipped. 

Our reply: We opted to focus on the five livestock species of major commercial relevance worldwide.

Some of the statements are without references, Proper references should be added to such sentences. 

Our reply: We carefully revised the entire MS and either added the missing references or deleted the passages.

Reviewer 3 Report

Summary

The aim of this review is to summarise the findings in the gut microbiota of the main animal production species using different methods of next-generation sequencing and to give an overall overview of the techniques, limitations and suggestions for future studies. In general, most of the studies have focused on faeces using amplicon metabarcoding, bacteria, cattle, and cosmopolitan breeds. While the number of studies is growing, more studies are needed with local breeds, covering the entire genome, function and other components of the gut microbiome.

Comments

Thank you so much for conducting this review and for providing suggestions for future studies. You have provided a good summary of the current state of research in this area. However, the paper would benefit from some modifications. Although three next-generation sequencing techniques are mentioned, the description of the findings is mainly based on one of them: Amplicon metabarcoding sequencing. While it is true that most of the studies are based on this method, it would be worth mentioning the main findings using the additional techniques, especially at the functional level.

On the other hand, it is well known that each segment of the GIT has different anatomic characteristics and microenvironments that cause differences in microbiota composition along the GI tract. However, the results are presented without distinction of the GI segment. At the phylum level, a general description can be given, but I would recommend a more specific description at the lower taxonomic levels, at least in rumen versus faecal microbiota. Also, in Tables S1-2, it would be useful to specify, GI segment, breed (as you mention this as one of the limitations of the current studies), the type of sample collected and the method employed. Also, without percentages provided, it is somewhat difficult to have a better idea about the abundance of the given taxa.

You have also mentioned that some studies have focused on methane production and environmental pollution. Members of the Archaea are the main methanogens, but nothing is mentioned about this domain.

It is also very important to mention, that several things can affect the results, from collection to storage, processing and bioinformatic analysis. Moreover, gut microbiota can be affected by several environmental factors as well as genetics and diet, adding complexity to the analysis. Hence the importance of standardisation for accuracy and reproducibility.  Moreover, the combination with other techniques such as metabolomics and proteomics (multi-omics sequencing) could provide more valuable information about the interaction of the host-microbiota-environment, which could be useful for future applications and interventions. Finally, these techniques are facing other challenges besides the cost, including the bioinformatic and statistical analysis.

Please bear in mind the nomenclature of bacteria, as some taxonomic levels are required to be printed in italics. 

Minor comments

Line 84: bacteria should be removed as it has been mentioned in the previous sentence.

Line 107: The results now can be presented in the form of OTUS, z-OTUS or ASVs.

Line 178: you mention faecal microbiota and give some examples, however, GIT location implies the collection of other types of samples, besides the faeces. This is confusing.

Line 185: It is a little bit confusing; do you mean that despite taxonomic composition differences, at the functional level, there are no differences?

Line 223: main drivers of what?

Line 235: What do you mean by GI traits? Do you mean segments?

Line 281: What were the findings of that study?

Line 374 and 378: different font has been used.

Author Response

REFEREE #3

Summary

The aim of this review is to summarise the findings in the gut microbiota of the main animal production species using different methods of next-generation sequencing and to give an overall overview of the techniques, limitations and suggestions for future studies. In general, most of the studies have focused on faeces using amplicon metabarcoding, bacteria, cattle, and cosmopolitan breeds. While the number of studies is growing, more studies are needed with local breeds, covering the entire genome, function and other components of the gut microbiome.

Our reply: We appreciate the reviewer's ability to capture the core of our MS in this succinct summary.

Comments

Thank you so much for conducting this review and for providing suggestions for future studies. You have provided a good summary of the current state of research in this area. However, the paper would benefit from some modifications. Although three next-generation sequencing techniques are mentioned, the description of the findings is mainly based on one of them: Amplicon metabarcoding sequencing. While it is true that most of the studies are based on this method, it would be worth mentioning the main findings using the additional techniques, especially at the functional level.

Our reply: We are appreciative that the reviewer found our work to be worthwhile. Please note that while we specifically mentioned the still small number of studies using methods other than 16rRNA amplicon metabarcoding as a specific drawback of the current state-of-the-art, we also mentioned the results obtained in the case of the ones carried out insofar as the three NGS approaches are concerned (e.g. 78, 102, 103, 106, 108, 134, 135, 137, 155).

On the other hand, it is well known that each segment of the GIT has different anatomic characteristics and microenvironments that cause differences in microbiota composition along the GI tract. However, the results are presented without distinction of the GI segment. At the phylum level, a general description can be given, but I would recommend a more specific description at the lower taxonomic levels, at least in rumen versus faecal microbiota. Also, in Tables S1-2, it would be useful to specify, GI segment, breed (as you mention this as one of the limitations of the current studies), the type of sample collected and the method employed. Also, without percentages provided, it is somewhat difficult to have a better idea about the abundance of the given taxa.

Our reply: The purpose of the tables and of this review in general is to give the readership a general overview of this topic while providing all the elements for obtaining more comprehensive information. We recognize that the content in Table S1 and S2 could be expanded, but we cannot condense all this information for all the studies mentioned there. Please, note that the most prevalent orders within the various GIT segments are listed in Figures 1 and 2.

You have also mentioned that some studies have focused on methane production and environmental pollution. Members of the Archaea are the main methanogens, but nothing is mentioned about this domain.

Our reply: We are aware of this drawback, but due to space restrictions (following the formatting for Review papers it should be brief), we decided to only briefly discuss it and concentrate on members of the Eubacteria (in fact there are no Archaea listed in the two Supplementary tables: we changed the text in their captions to specify we are focusing on Eubacteria).

It is also very important to mention, that several things can affect the results, from collection to storage, processing and bioinformatic analysis. Moreover, gut microbiota can be affected by several environmental factors as well as genetics and diet, adding complexity to the analysis. Hence the importance of standardisation for accuracy and reproducibility. Moreover, the combination with other techniques such as metabolomics and proteomics (multi-omics sequencing) could provide more valuable information about the interaction of the host-microbiota-environment, which could be useful for future applications and interventions. Finally, these techniques are facing other challenges besides the cost, including the bioinformatic and statistical analysis.

Our reply: This is an excellent point. In order to include these components, further text was added to the Conclusions.

Please bear in mind the nomenclature of bacteria, as some taxonomic levels are required to be printed in italics. 

Our reply: Yes, we meticulously edited the MS to ensure that all the genera were listed in italics. This and other formatting aspects may have been lost during the creation of the PDF.

Minor comments

Line 84: bacteria should be removed as it has been mentioned in the previous sentence.

Our reply: Done.

Line 178: you mention faecal microbiota and give some examples, however, GIT location implies the collection of other types of samples, besides the faeces. This is confusing.

Our reply: We removed the word fecal in this passage to avoid any confusion.

Line 185: It is a little bit confusing; do you mean that despite taxonomic composition differences, at the functional level, there are no differences?

Our reply: Not quite. As implied in the title of the work quoted in this line, we said that the suite of microbial taxa may differ yet still serve the same purpose:

Taxis, T.M.; Wolff, S.; Gregg, S.J.; Minton, N.O.; Zhang, C.; Dai, J.; Schnabel, R.D.; Taylor, J.F.; Kerley, M.S.; Pires, J.C.; et al. The players may change but the game remains: network analyses of ruminal microbiomes suggest taxonomic differences mask functional similarity. Nucleic Acids Res. 2015, 43, 9600-9612, doi:10.1093/nar/gkv973.

Line 223: main drivers of what?

Our reply: We added the words of feed efficiency” to improve clarity.

Line 235: What do you mean by GI traits? Do you mean segments?

Our reply: Right, we preferred to use "locations" to keep with the nomenclature used up to that point.

Line 281: What were the findings of that study?

Our reply: The findings of that study are previously briefly discussed in paragraph 4.1.3, where we note that both sheep and cows have shown a clear difference in microbial composition among GIT locations.

Line 374 and 378: different font has been used.

Our reply: To ensure that the same typeface is utilized across the MS, we double-checked it. As was already indicated above, it appears that there was an issue with the generation of the pdf.